# Clinical and Biochemical Characterization of Hereditary ATTR Amyloidosis Caused by a Novel Transthyretin Variant V121A (p.V141A)

**DOI:** 10.3390/ijms26104659

**Published:** 2025-05-13

**Authors:** Tsuneaki Yoshinaga, Yuuki Yoshioka, Felix J. Tsai, Luke Nelson, Ming Cheng, Ryota Ito, Satoshi Fujita, Eri Ishikawa, Fuyuki Kametani, Ryuzi Aoyagi, Takahiro Okumura, Toyoaki Murohara, Masahide Yazaki, Yoshiki Sekijima

**Affiliations:** 1Department of Chemistry, The Scripps Research Institute, 10550 North Torrey Pines Road, La Jolla, CA 92037, USA; ftsai98@uw.edu (F.J.T.); luketn@hawaii.edu (L.N.); chengming@simm.ac.cn (M.C.); 2Department of Neurology and Rheumatology, Shinshu University, Matsumoto 390-8621, Japan; sekijima@shinshu-u.ac.jp; 3Institute for Biomedical Sciences, Shinshu University, Matsumoto 390-8621, Japan; kametani-fy@igakuken.or.jp (F.K.); mayazaki@shinshu-u.ac.jp (M.Y.); 4Department of Nephrology, Tachikawa General Hospital, Nagaoka 940-0831, Japan; yuukiy@chiba-u.jp (Y.Y.); ry-aoyagi181@tatikawa.or.jp (R.A.); 5Department of Cardiology, Nagoya University Graduate School of Medicine, Nagoya 466-8560, Japan; ir009013@med.nagoya-u.ac.jp (R.I.); murohara@med.nagoya-u.ac.jp (T.M.); 6Department of Cardiology, Tachikawa General Hospital, Nagaoka 940-0831, Japan; sa-fujita121@tatikawa.or.jp; 7Division of Instrumental Research, Research Center for Advanced Science and Technology, Shinshu University, Matsumoto 390-8621, Japan; eishikawa@shinshu-u.ac.jp; 8Department of Dementia and Higher Brain Function, Tokyo Metropolitan Institute of Medical Science, Tokyo 156-0057, Japan

**Keywords:** amyloid, transthyretin, hereditary transthyretin amyloidosis, ATTR cardiomyopathy

## Abstract

Over 150 transthyretin (TTR) mutations have been identified in hereditary transthyretin (ATTRv) amyloidosis, and new TTR variants have recently emerged. However, the pathogenicity of several new variants remains unclear, making it important to elucidate the differences between amyloidogenic and wild-type TTR. In this study, we report a novel TTR variant (V121A) identified in two unrelated amyloidosis patients aged > 60 years who developed cardiomyopathy. We evaluated the detailed biochemical features of this TTR variant to confirm its amyloidogenicity using plasma samples from these patients and recombinant TTR proteins. While the V121A TTR variant has a similar ability to assemble into a tetramer as wild-type TTR, it aggregates more readily over a wide potential hydrogen range than wild-type TTR. Additionally, the V121A variant is highly prone to dissociation and resistant to binding with known TTR tetramer stabilizers. Clinical and biochemical data suggest that this novel variant is clearly pathogenic, is highly prone to dissociation and aggregation, and is associated with the development of late-onset amyloid cardiomyopathy. Interestingly, amyloid fibril formation due to this variant may not be affected by known TTR stabilizers.

## 1. Introduction

Transthyretin amyloidosis (ATTR amyloidosis) is a life-threatening disease characterized by the accumulation of amyloid fibrils composed of transthyretin (TTR) [1,2]. TTR is a 55 kDa tetramer consisting of four β-sheet-rich subunits, each comprising 127 amino acid residues [3]. TTR is primarily synthesized in the liver; however, smaller quantities are produced in the choroidal plexus and retina. The TTR tetramer binds and transports thyroxine- and holoretinal-binding proteins. Binding to either ligand slows TTR tetramer dissociation, which is the rate-limiting step of TTR protein aggregation [3,4,5]. Following this rate-limiting tetramer dissociation and partial monomer denaturation, TTR can aggregate into numerous non-native structures, including cross-β-sheet amyloid fibrils, and the soluble non-native TTR structures appear to be the main drivers of degenerative phenotype [6]. The wild-type TTR (WT-TTR) tetramer is kinetically stable, and it dissociates with a half-life on the order of 1 day in vitro at 25 °C and 2–3 days ex vivo in the blood [7,8]. The misfolding of wild-type TTR results in wild-type ATTR amyloidosis (ATTRwt amyloidosis), which presents as an acquired amyloid disease in the elderly [9].

Hereditary ATTR amyloidosis (ATTRv amyloidosis) associated with variant TTR has historically been an intractable disease. TTR heterotetramers composed of both mutant TTR and WT-TTR caused faster dissociation into monomer subunits and aggregation for amyloid fibril formation. Several therapeutic strategies have been established to slow or halt disease progression in patients with ATTRv amyloidosis. One of the main strategies is the stabilization of the tetramer structure with small-molecule ligands, such as diflunisal [10], tafamidis [11], tolcapone [12], and acoramidis [13]. These molecules, termed kinetic stabilizers, preferentially bind to the thyroxine-binding sites of TTR. Binding to the tetramer can slow its dissociation into monomers [14]. In addition to these tetramer stabilizers, TTR silencer drugs, including patisiran [15], vutrisiran [16], inotersen [17], and eplontersen [18], which reduce the hepatic synthesis of TTR, have also been used in patients with ATTRv amyloidosis.

To date, over 150 TTR variants have been identified in ATTRv amyloidosis. However, the pathogenicity of some of these variants is not fully understood, and it is important to elucidate the differences in amyloidogenicity compared to wild-type TTR. Here, we report the cases of two unrelated patients with late-onset amyloid cardiomyopathy. Both patients carried a novel TTR variant (V121A), and we evaluated the detailed biochemical features of this new TTR variant and compared them to those of wild-type TTR. We also investigated the pharmacological effect of known TTR stabilizers on this TTR variant.

## 2. Patients

### Two Unrelated Patients with ATTRv and a Novel TTR Variant V121A (p.V141A)

The clinical and laboratory data of the two patients are summarized in Table 1. Additionally, brief presentations of the cases are provided. ATTR amyloidosis was diagnosed by endomyocardial biopsy, and a novel mutation (heterozygous c.422T > C, p.V141A) in the TTR gene was confirmed by DNA analysis (Appendix A). Informed consent was obtained from both patients, and the study was conducted in accordance with the ethical guidelines of the 2013 Declaration of Helsinki.

Patient 1 was a 67-year-old Japanese male with no family history of amyloidosis (Figure 1A). He noticed leg edema at the age of 62 years, and cardiomegaly was observed on echocardiography. Subsequently, he showed signs of heart failure at the age of 67 years, with an enlarged heart shadow on chest radiography (Figure 1B). Simultaneously, he was diagnosed with lumbar spinal canal stenosis in the department of orthopedic surgery. Clear cardiac uptake was observed on 99mTc-PYP scintigraphy [H/CL early 1.8 (1 h), delay 1.5 (3 h)] (Figure 1C), leading to suspicion of ATTR amyloidosis based on red flag signs of cardiac hypertrophy (Figure 1D) and spinal canal stenosis. Therefore, an endomyocardial biopsy revealed amyloid deposits specifically immunolabeled with anti-TTR antibodies (Figure 1E). DNA analysis showed a heterozygous c.422T > C, p.V141A mutation in the TTR gene. After the diagnosis, the patient was treated with RNAi every 3 months. The patient was followed up for 12 months and experienced no major side effects.

Patient 2 was a 71-year-old Japanese male with a history of hypertension and hyperuricemia. The patient had no family history of amyloidosis (Figure 2A). He had slightly elevated serum creatinine levels since the age of 67 years and signs of heart failure (NYHA functional class III) at 70 years of age. Diuretic therapy was initiated; however, his weight did not decrease, and renal function worsened further. Enlargement of the cardiac shadow with bilateral pleural fluid was observed on the chest radiograph (Figure 2B). The patient was transferred to the department of nephrology for emergency hemodialysis. There were no overt signs of neuropathy, including carpal tunnel syndrome. However, amyloidosis was strongly suspected due to myocardial wall thickening (Figure 2C), and endomyocardial biopsy showed amyloid deposits specifically immunolabeled with an anti-TTR antibody (Figure 2D). DNA analysis revealed a heterozygous c.422T > C, p.V141A mutation in the TTR gene (Appendix A). Two months later, the patient was readmitted to the hospital because of renal dysfunction, decreased appetite, weight loss, and dyspnea. Hemodialysis was resumed; however, the patient developed respiratory failure. Despite conventional treatment, the patient died of heart failure 4 months after the first admission.

Postmortem examination revealed a heart weight of 800 g (normal Japanese heart weight, <300 g). Significant amyloid deposition with the loss of myocardial fibers was observed at the interstitial lesions in the heart. However, there was less amyloid deposition in the kidneys and peripheral and central nervous system. Our LC-MS/MS data showed that the amyloid deposited in the myocardial specimen contained V121A (p.V141A) peptide (Appendix A and Appendix A).

## 3. Results

### 3.1. Clinical Features of Two Patients

Regarding the common clinical features of these two patients, they developed heart failure in their 60s, with wall thickness findings on echocardiography and high serum cardiac marker levels (BNP and NT-proBNP). There were no signs of carpal tunnel syndrome, autonomic neuropathy, or ocular or CNS manifestations. Their serum transthyretin levels were low (Table 1). However, Patient 1 suffered from spinal canal stenosis and sensory disturbances in his foot, and it was unclear whether these symptoms were due to neuropathy or spinal canal stenosis.

### 3.2. Evaluation of Pathogenicity Associated with This Novel TTR Variant

#### 3.2.1. In Silico Analysis of V121A

The uniallelic V121A (p.V141A) variant has not been previously reported or described in several databases, such as the Human Gene Mutation Database Professional and ClinVar. An in silico prediction program, CADD (https://cadd.gs.washington.edu/ accessed on 8 December 2017), predicted this variant to be disease-causing, with a score of 21.9 (>20). A score of 20 or higher is considered disease specific. This method uses ensemble learning to produce a score by combining various previously used predictive scores for pathogenicity, such as SIFT and PolyPhen-2. The V at the 121 residue of TTR is conserved among chimpanzees, rhesus macaques, mice, chickens, and sea bream (Appendix A).

#### 3.2.2. Analyses of Tetramer Formation Ability in the V121A(p.V141A) Mutation

Normally, TTR forms tetramers; however, in several TTR mutations [19,20], TTR dissociates into dimers and monomers instead of remaining as a tetramer. Therefore, we hypothesized that this mutation could easily cause dissociation and instability because the V121 residue is located in the dimer (Figure 3A). To test this hypothesis, we first examined the ability of V121A to form a tetramer. The recombinant V121A (p.V141A) TTR was subjected to native gel electrophoresis, which revealed the presence of a distinct tetrameric band under native conditions, indicating a tetrameric state. In the native-PAGE gel, the density of the V121A (p.V141A) band was similar to that of the WT-TTR, suggesting that the mutant TTR can also form a tetramer (Figure 3B, Appendix A). When varying the concentrations of V121A (p.V141A) from 1 μM to 10 μM, it consistently existed as a tetramer in the native-PAGE gel (Appendix A). In addition, no clear change was observed in the density of the tetramer bands of V121A conjugated with A2, a small fluorescent molecule that selectively binds to TTR (Figure 3B). DLS results are generally indicators of the size of the most abundant protein. The average radius of V121A (p.V141A) was similar to that of WT-TTR. Both sets of data showed that V121A (p.V141A) TTR could form a tetramer (Figure 3C). A subunit exchange assay (His)^6^-V121A transthyretin vs. (His)^6^-(FLAG)^2^-WT-TTR) showed five peaks resulting from UPLC chromatography (Figure 3D), suggesting that V121A TTR functions as a tetramer (Figure 3D).

#### 3.2.3. V121A (p.V141A) Aggregates Under Acidic Conditions Despite the Presence of Kinetic Stabilizers

The aggregation assay showed various TTR aggregations at 37 °C for 72 h at the desired pH. V121A (p.V141A) was broadly aggregation-prone under mildly acidic conditions (pH 4.0–5.0) (Figure 4A,B). The level of aggregation was notably different from that of WT-TTR. The V121A (p.V141A) tetramer with the stabilizer at pH 4.4 is shown in Figure 4B. Many small-molecule stabilizers, such as diflunisal and tafamidis, have been reported to inhibit TTR fibril formation. We studied the inhibitory effects of these stabilizers on amyloid formation in the V121A variant. While increasing concentrations of tafamidis (Figure 4C) and diflunisal (Figure 4D) dramatically reduced aggregation propensity at concentrations below 15 μM, we found that the inhibitory effects of tafamidis and diflunisal on V121A (p.V141A) fibril formation were less significant, with neither stabilizer reducing aggregation to less than 50% of vehicle treatment even at high concentrations (28μM tafamidis, 280 μM diflunisal). While acoramidis treatment reduced aggregation propensity at somewhat lower concentrations than tafamidis or diflunisal, it remained higher for V121A TTR than WT (Figure 4E). Isothermal titration calorimetry (ITC) of diflunisal with WT and V121A showed that WT had approximately 92 times higher affinity for diflunisal compared to V121A (Figure 4F,G).

#### 3.2.4. Proportion Ratio of TTRs in Serum Variant Mutants Including V121A (p.V141A)

Several studies have been conducted on the stability of TTR proteins and their efficiency of secretion into the bloodstream [19,20]. They showed that unstable proteins were less efficiently secreted into the bloodstream. Owing to its high aggregation propensity, we hypothesized that the V121A mutation might also result in poor secretion efficiency. In our study, the variant TTR (V121A) ratio of the total TTR was 26.3% ± 0.01%. This ratio was relatively low compared to other variants for cardiac amyloidosis and mixed phenotype, including T59K (48.6% ± 0.00%) [22], V122I (44.5% ± 0.01%,) [23], A97S (43.1% ± 0.01%) [24], F64L (43.9% ± 0.01%) [25], and V30L (50.5% ± 0.01%) [26] (Figure 5). However, the ratio in V30G [27], which is associated with leptomeningeal amyloidosis, was almost similar (26.6% ± 0.02%) to that of the V121A variant.

## 4. Discussion

We demonstrated amyloidogenesis in the clinical presentation and the biochemical analysis of the V121A variant. Two patients presented with late-onset cardiomyopathy, with one also having spinal canal stenosis. Notably, one patient died within 4 months of onset. The most common TTR mutant, V122I (p.V142I), is notably prevalent among African-Americans (3.0–3.9%) [28] resulting in a median survival of 25.6 months after diagnosis [23], which is slightly shorter than that in ATTRwt amyloidosis (43–75 months) [29]. In contrast, Patient 2 in this study died 4 months after the diagnosis, and their clinical course was similar to that of patients with heart failure due to light-chain amyloidosis, where the median survival of untreated patients is less than 6 months [30]. The prognosis of ATTRv amyloidosis with V121A variant may be worse than ATTRwt amyloidosis, although there is room for further investigation because of the small number of cases.

Regarding the pathogenicity associated with amyloid formation due to this novel mutation, V121A, residue V121 is located on the H-strand and at the dimer–dimer interfaces (Figure 3A), suggesting that this mutation may strongly affect the dimer structure of the TTR molecule and have clear pathogenicity. Mutations near V121 (such as delta 122 and V122I) are all pathogenic.

While the acid denaturation pathway of TTR has been extensively studied to understand amyloid fibril formation, the V121A protein was clearly aggregation-prone at less acidic conditions in the acid denaturation pathway compared to WT-TTR. In the aggregation reaction with recombinant TTR proteins, TTR proteins with known pathogenic mutations such as A25T, D18G, and L55P exhibit a wide range (pH 3.2–6) of acidic to basic aggregation in the acid denaturation pathway, while the aggregation range of benign mutations and WT tends to be more limited to acidic conditions (pH 3.2–4.2) [19]. This suggests that recombinant V121A, similar to these pathogenic mutations, has a high capacity for amyloid fibril formation.

Data from the aggregation inhibition assay at pH 4.4 using tetramer stabilizers showed that tafamidis and diflunisal did not effectively inhibit amyloid fibril formation in the V121A TTR protein. Even at the highest physiological stabilizer concentrations reported for buffers (28 μM tafamidis and 280 μM diflunisal), both tafamidis and diflunisal showed only up to 50% inhibition. In contrast, acoramidis inhibited V121A protein aggregation relatively well. These results were consistent with data from the aggregation inhibition assay, as the binding of V121A to diflunisal in isothermal titration calorimetry showed an approximately 100-fold decrease in affinity compared with that of WT to diflunisal.

To date, some TTR mutations have been shown to confer resistance to tetramer stabilizers. For example, the A25T mutation affects the tetramer–monomer equilibrium, and TTRs with this mutation are resistant to binding because they tend to be monomer-rich [31]. The Glu51_Ser52dup mutation binds to the tetramer stabilizer diflunisal but is still susceptible to protease-terminated aggregation [32]. The fact that V121A (p.V141A) may also be unresponsive to tetramer stabilizers, as shown in this study, is important for therapeutic drug options.

Patients with the V121A mutation have lower serum TTR levels, and the mutation is associated with a decreased ratio of 23% of all TTR. Patients with hereditary ATTR amyloidosis are reported to have decreased serum TTR concentrations after the onset of amyloidosis, and the production and survival ratio of wild-type TTR to mutant TTR is generally about 1:1 [24,33]. Low serum variant TTR levels have been observed in a small number of patients with highly destabilizing mutations (such as D18G and A25T) [19,23,26]. This is consistent with the retention of unstable TTRs in the endoplasmic reticulum observed in cell biology studies and with proteasomal degradation, leading to low serum concentrations of these unstable TTRs. In this patient, the percentage of V121A TTRs in the serum was very low. However, the constitutive percentage of mutant TTR in tissue amyloids exceeded 50% [34], suggesting that V121A TTR selectively aggregates in tissues, even at low serum concentrations, when the V121A peptide affects deposition.

We have several limitations; our data are of a small sample size. Also, we lack longitudinal data. Future research should prioritize larger cohorts, diverse populations, and deeper investigations into molecular mechanisms and treatment strategies to validate this variant.

## 5. Methods

### 5.1. Evaluation of Pathogenicity Associated with This Novel TTR Variant

#### 5.1.1. Survey in Public Databases and Genetic Analysis In Silico

The V121A mutation was analyzed in databases of disease-causing variants [Human Gene Mutation Database Professional (http://www.hgmd.org/ accessed on 8 December 2017) and ClinVar (https://www.ncbi.nlm.nih.gov/clinvar/ accessed on 8 December 2017)], and healthy individual databases [The 1000 Genomes Project (http://www.internationalgenome.org/ accessed on 8 December 2017) and Exome Aggregation Consortium (https://gnomad.broadinstitute.org/ accessed on 8 December 2017)]. We examined the conservation of V121 residues in TTR among heterologous animals. Bioinformatics tools [SIFT (https://sift.bii.a-star.edu.sg/ accessed on 8 December 2017), PolyPhen2 (http://genetics.bwh.harvard.edu/pph2/ accessed on 8 December 2017), Mutation Taster (http://www.mutationtaster.org/ accessed on 8 December 2017), and CADD (http://cadd.gs.washington.edu/ accessed on 8 December 2017)] were used to assess the possible effects of the novel mutants on TTR structure and function (Appendix A).

#### 5.1.2. 3D Structural Prediction

The X-ray crystal structure of TTR is available in the Protein Data Bank (PDB). We used the SWISS-MODEL (http://swissmodel.expasy.org/interactive/ accessed on 25 June 2020) to generate a 3D structural model of V121A (p.V141A). The models were saved in the PDB format and visualized using the PyMOL tool (http://pymol.sourceforge.net/ accessed on 25 June 2020).

#### 5.1.3. Expression and Purification of Recombinant TTR Protein

Plasmids encoding N-terminal (His)^6^-V121A (p.V141A) were transformed into BL21(DE3) Escherichia coli cells (Appendix A). These constructs contained C10A to remove potential artifacts caused by thiol oxidation of cysteine 10.

For the pH aggregation assay, the V121A (p.V141A) TTR protein and WT-TTR were used without the N-terminal (His)^6^ or C10A (Appendix A). Expression and purification were performed as previously described [4].

#### 5.1.4. Non-Denatured Electrophoresis (Native PAGE)

Recombinant TTR [both V121A (p.V141A) and WT-TTR] was analyzed by Native PAGE. A 4–16% Bis-Tris Gel was used with a native running buffer (50 mM Bis-Tris, 50 mM Tricine, pH 6.8). Samples were diluted with native sample buffer (62.5 mM Tris-HCl, pH 6.8, 40% glycerol, 0.01% bromophenol blue) and run using a Tris-Glycine native running buffer at 180 V for 90 min. The gel was then stained with Coomassie Blue for 30 min, after which it was rinsed with dH_2_O. Protein bands were visualized and captured using a ChemiDoc system (Bio-Rad, Hercules, CA, USA). We recruited A2, a small fluorescent molecule that selectively binds to TTR and displays the same signal intensity on the gel. Prior to gel electrophoresis analysis, the recombinant V121A (p.V141A) TTR was incubated with A2 (final concentration = 500 μM) for 3 h. This ensured complete covalent labeling of the Lys15 residues in two of the four TTR subunits within each native tetramer [35].

#### 5.1.5. Measurement of Particle Size (Dynamic Light Scattering)

Dynamic light scattering (DLS) (DynaPro NanoStar; Waters, Milford, Massachusetts, USA) measures the hydrodynamic radii of particles in a solution by monitoring the fluctuations in the intensity of scattered light due to Brownian motion. This technique was used to determine the radii of the recombinant WT and V121A (p.V141A) TTR. The data were analyzed using DYNAMICS (version 7.6.0) software. To obtain the hydrodynamic radii, an autocorrelation function, Regularization, was employed. The following values were measured for each TTR sample: percentage intensity, percentage mass, and percentage number. To assess the native size of TTRs in comparison to WT-TTR, the percentage number distribution typically indicates the size of the most prevalent protein.

#### 5.1.6. Endpoint of Subunit Exchange Assay

Recombinant TTRs [(His)^6^-C10A-V121A] and [(His)^6^-(FLAG)^2^-WT TTR] were mixed at a concentration of 5 μM over a period of 0–5 days, followed by analysis using ion exchange chromatography at two time points (days 0 and 5). Initial populations of homotetramers were mixed to create heterotetramers incorporating between zero and four (FLAG)^2^-TTR subunits. The samples were injected into the UPLC system to determine the endpoint of the subunit exchange.

#### 5.1.7. TTR Aggregation Measured by Turbidity

TTR (1.0 mg/mL) in 20 mM phosphate buffer with 100 mM KCl (pH 7.0) was diluted 1:1 with 200 mM acetate buffer (100 mM KCl and 1 mM EDTA) to adjust to the desired pH. The solutions were subjected to denaturation stress and incubated at 37 °C for 72 h, after which the suspensions were vortexed and optical density was measured at 400 nm. One hundred microliters of the solution were transferred into 96-well transparent plates in triplicate. The turbidity of the solutions was recorded at 400 nm using a microplate reader (SpectraMax M2; Molecular Devices, San Jose, CA, USA). The average optical density of the same solvent without protein was subtracted for each experimental sample. The data presented correspond to the average values from one experiment, and the error bars represent the standard deviation.

#### 5.1.8. Evaluation of TTR Aggregation with Known TTR Stabilizers

TTR (7.2 μM) in phosphate buffer (20 mM sodium phosphate, 100 mM KCl, pH 7.4) was preincubated with known TTR stabilizers, including tafamidis, diflunisal, acoramidis, and A2 at various concentrations at 37 °C for 0.5 h before shifting the pH to the desired acidic conditions. This was achieved by adding an acidic buffer (100 mM KCl, 1 mM EDTA, pH 4.4). All samples were incubated at 37 °C for 72 h, and the extent of fibril formation was measured by OD at 400 nm. The turbidities of the solutions at 400 nm were recorded using a microplate reader (SpectraMax M2; Molecular Devices, San Jose, CA, USA). Aggregation in the presence of the solvent alone (DMSO) was defined as 100% fibril formation.

#### 5.1.9. Isothermal Titration Calorimetry

Thermodynamic parameters describing the interaction between TTR and diflunisal were determined by Isothermal titration calorimetry using a MicroCal Auto-iTC200 Calorimeter (MicroCal; Malvern Panalytical, Malvern, Worcestershire, UK). V121A and WT-TTR at 25 μM located in the calorimetric cell were titrated against diflunisal at 500 μM in the injection syringe in 137 mM NaCl, 100 mM KCl, 10 mM Na_2_HPO_4_, 1.8 mM KH_2_PO_4_, 1 mM EDTA, and 2.5% DMSO, at 25 °C and pH 7.0. A stirring speed of 750 rpm and 2 μL injections were programmed, with consecutive injections separated by 150 s to allow the calorimetric signal (thermal power) to return to baseline. Two replicates were performed for each protein, and the experimental data were analyzed using a general model for a protein with sequential binding sites implemented in ORIGIN 7.0 (OriginLab, Northampton, MA, USA). Appropriate control blank experiments were also conducted to test for unwanted heat artifacts or nonspecific phenomena.

#### 5.1.10. Proportion of Variant TTR to Wild-Type TTR in Serum

The ratios of variant to wild-type TTR in the plasma of seven patients with ATTRv amyloidosis and seven types of TTR variants were examined: T59K (p.T79K), V122I (p.V142I), A97S (p.V117S), F64L (p.F84L), V30L (p.V50L), V30G (p.V50G), and V121A (p.V141A) (Appendix A).

TTR molecules extracted from the subjects’ serum were analyzed using a liquid chromatography–mass spectrometry detector (LC-MSD; Agilent, Santa Clara, CA, USA), following the established protocol for immunoprecipitation–mass spectrometry (IP-mass analysis) [36]. Twenty-microliter serum samples were incubated with 20 μL of rabbit anti-human TTR antiserum (Dako, Glostrup, Denmark) for 12 h at 4 °C, followed by centrifugation. The precipitates were washed twice with 200 μL of saline and then once with 200 μL of distilled water. The precipitates were treated with 20 μL of 0.1 M dithiothreitol at room temperature for 15 min. Samples were lyophilized overnight and resuspended in 10 μL of 2:1 0.1% TFA/ACN. Samples were loaded into a Waters XTERRA MS C8 column (3.5 μm, 1.6 × 100 mm) using an Agilent LC-MSD, employing a gradient of 5 to 50% solution B in 4 min. Subsequently, the gradient was held at 50% solution B for 2 min before reaching 95% solution B over 1 min, all at a flow rate of 0.5 mL/min (solution A: 0.1% formic acid in water; solution B: 0.1% formic acid in acetonitrile). Peak apex mass reflects wild and amino acid substitution (Appendix A)

Ion intensity was quantified from the total ion chromatogram detected by LC-MSD. Each intensity was calculated as the V121A (p.V141A) intensity divided by the total [V121A (p.V141A) and WT-TTR] intensity (Appendix A).

## 6. Conclusions

We report two unrelated patients with ATTRv amyloidosis and a novel V121A (p.V141A) variant in the TTR gene. Both patients exhibited late-onset cardiomyopathy. In addition, our biochemical data clearly showed that the V121A variant is highly dissociation-prone and pathogenic. Notably, this mutant may be resistant to binding with the FDA-approved stabilizers tafamidis and diflunisal; thus, patients with this type of TTR mutation should be treated with alternative strategies, such as TTR silencing.

## Figures and Tables

**Figure 1 ijms-26-04659-f001:**
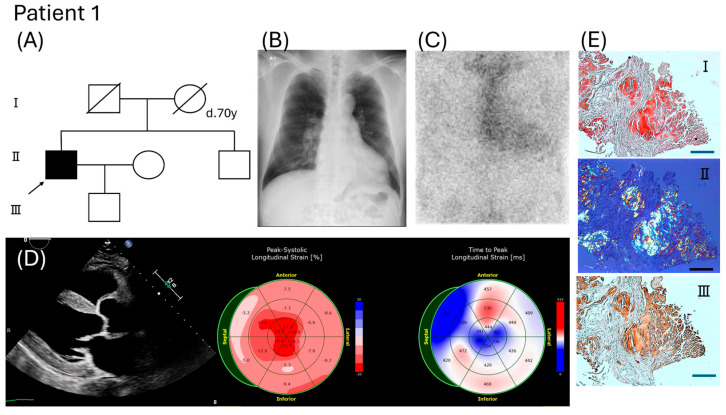
Clinical features of Patient 1. (**A**) Family pedigree. An arrow indicates the proband. (**B**) Chest radiography at age 67 showed enlargement of the cardiac shadow (CTR 57%). (**C**) Tc−99m−PYP scintigraphy revealed grade 3 myocardial uptake, and the heart−to−contralateral ratio was 1.783 (1 h). (**D**) Left ventricular long-axis view of two−dimensional echocardiography, speckle−tracking echocardiography. (**E**) Myocardial biopsy specimens for Congo red staining (**I**) (**II**, polar) and immunohistochemistry with anti−TTR antibody (**III**). Bar = 100 μm.

**Figure 2 ijms-26-04659-f002:**
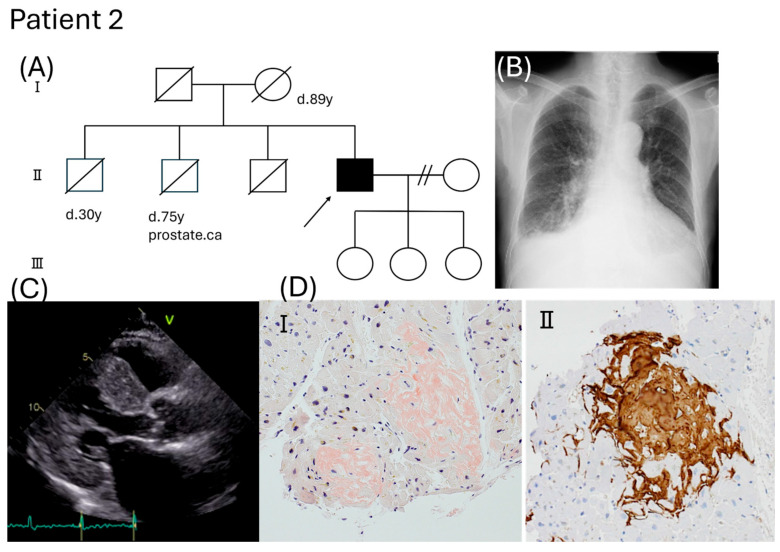
Clinical features of Patient 2. (**A**) Family pedigree. An arrow indicates the proband. (**B**) Left ventricular long−axis view of two-dimensional echocardiography. (**C**) Chest radiography showed enlargement of the cardiac shadow (CTR 63%) with bilateral effusion. (**D**) Myocardial biopsy specimens for Congo red staining (**I**) and immunohistochemistry with anti−TTR antibody (**II**).

**Figure 3 ijms-26-04659-f003:**
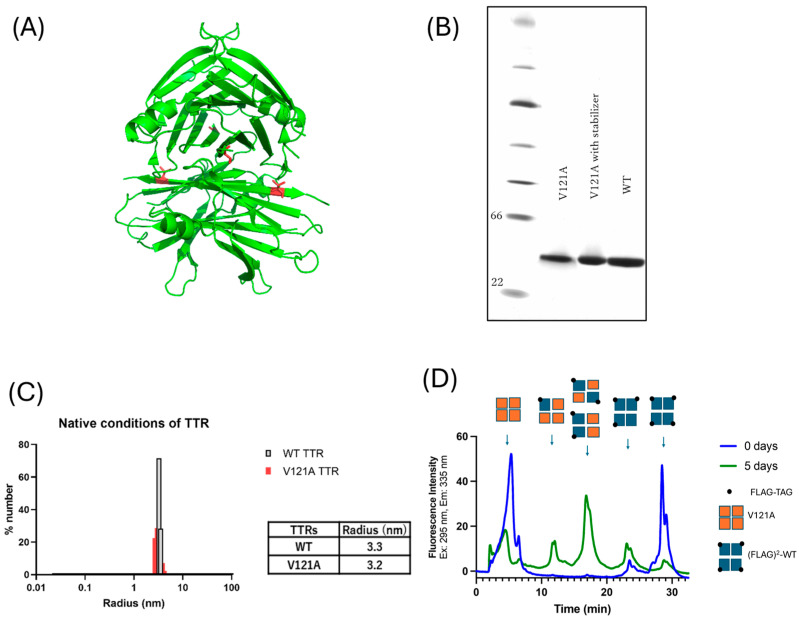
(**A**) 3D model of V121A-TTR based on the X-ray crystallographic structure WT-TTR (PDB code 1F41 [21]). The weak dimer–dimer interface is initially separated from the tetramer. Red dots indicate the V121A residue in the beta-sheet (H). The surface of the dimer–dimer interface is important for retaining two T4 binding sites and the tetramer structure. For analysis of native state TTR, we utilized (His)^6^-V121A transthyretin and (His)^6^-(FLAG)^2^-Wild-type transthyretin (Appendix A). (**B**) Native PAGE. The density of V121A showed similarity to the tetramer signal of WT-TTR. Moreover, the density of V121A with the stabilizer (A2) was the same as that of WT-TTR. (**C**) Dynamic light scattering detected the distribution of TTR proteins. The percentage number distribution is generally an indication of the size of the most abundant protein. The average radius indicates that V121A is similar to WT-TTR, which means both are tetramers. (**D**) Subunit exchange of (His)^6^-V121A transthyretin vs (His)^6^-(FLAG)^2^-WT-TTR). After 5 days of incubation, there were five peaks, with V121A serving as a tetramer.

**Figure 4 ijms-26-04659-f004:**
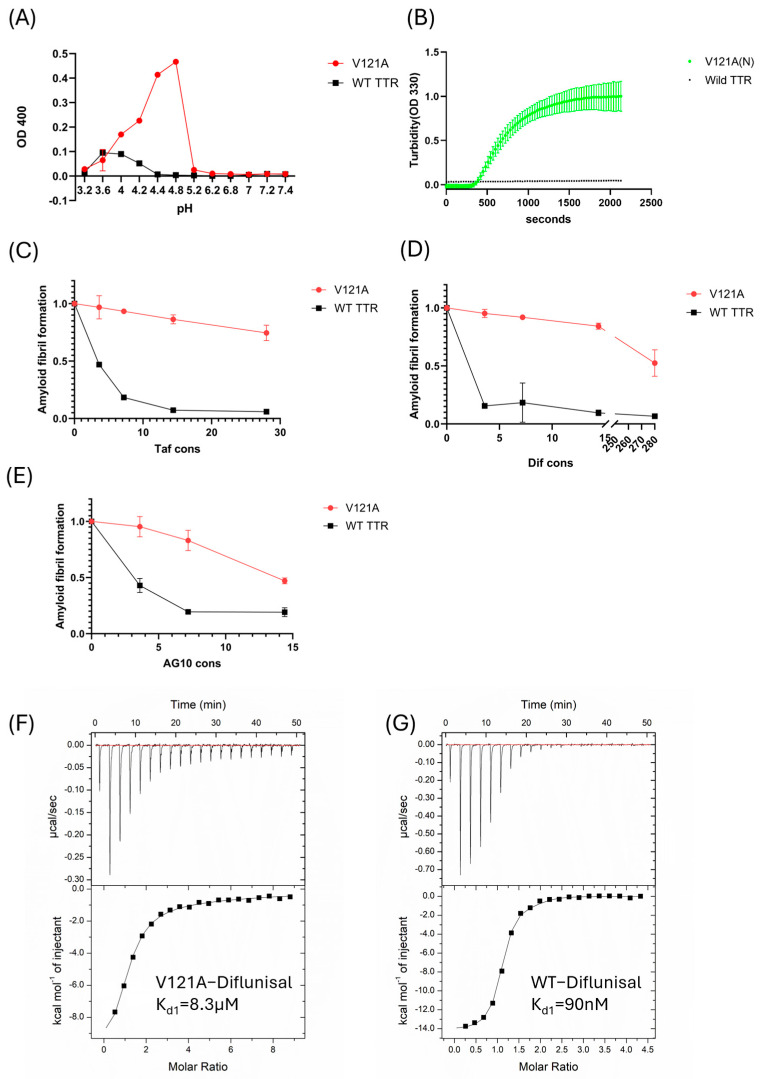
V121A TTR protein aggregation and small compounds for inhibiting aggregation in acidic conditions. For aggregation studies, we utilized (non-tag) V121A transthyretin and (non-tag) WT TTR (Appendix A). (**A**) The aggregation assay shows a tendency for aggregation at 37 °C at the desired pH for 72 h. The red line indicates V121A TTR, and the black line indicates WT-TTR. The bars represent the SEM. V121A shows a tendency for aggregation in mildly acidic (pH 4.0–5.0) conditions. (**B**) Kinetics of aggregation and fibril formation for each TTR. TTR mutants show significantly higher turbidity over a time course (30 s each period) in acidic conditions compared to WT-TTR. Aggregation of TTRs was induced via solution at pH 4.4 under the same conditions at 37 °C without agitation. Data are shown for two representative results (**C**–**E**) Tetramer stabilizers (tafamidis, diflunisal, and acoramidis) with V121A protein and WT protein. The highest physiological concentrations for tafamidis and diflunisal were 28 μM and 280 μM, respectively. The TTR concentration was 3.6 μM. All samples were incubated at 37 °C for 72 h, and the extent of fibril formation was measured using OD400. Aggregation of WT-TTR and V121A with DMSO as the vehicle control was defined as 100% fibril formation. (**F**,**G**) Isothermal titration calorimetry (ITC) of diflunisal with wild-type TTR and V121A. The dissociation constant of diflunisal with TTR was noted (TTR concentration 25 μM).

**Figure 5 ijms-26-04659-f005:**
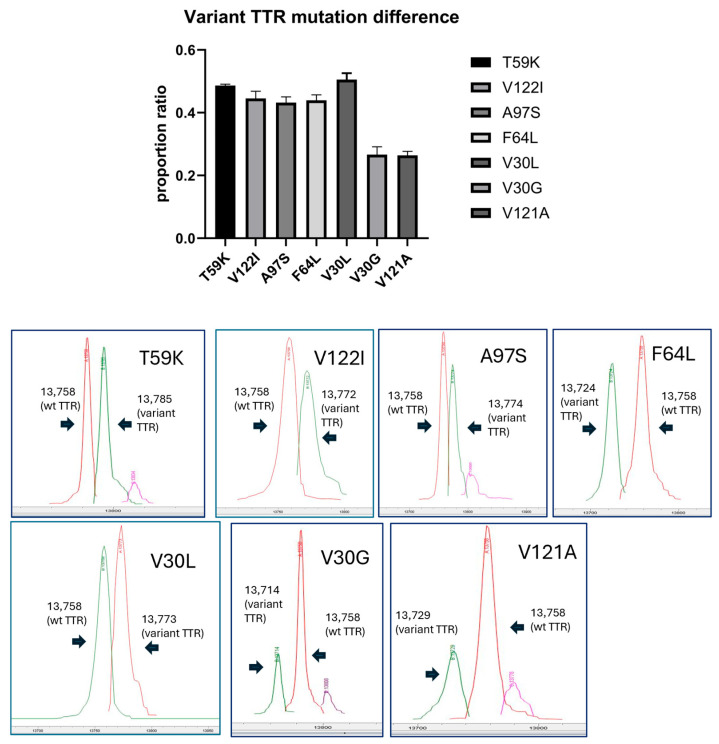
Proportion of variant TTR in various TTR mutations in serum using LC-MSD. V121A TTR comprised 26.3% ± 0.01 of total serum TTR, compared to other variants, with a similar ratio to the leptomeningeal type (V30G). Both wild-type TTR and variant TTRs were detected as follows: the mass detected using LC-MSD was 13,758 for wild-type TTR and 13,785 (T59K), 13,772 (V122I), 13,774 (A97S), 13,724 (F64L), 13,773 (V30L), 13,714 (V30G), and 13,729 (V121A) for other variants.

**Table 1 ijms-26-04659-t001:** Clinical features and laboratory data of two patients with V121A(p.V141A).

	Patient	
	1	2
Age of diagnosis (sex)	67 (M)	71 (M)
Age of onset	62	70
Ethnicity	Japanese	Japanese
Positive family history	none	none
Cardiac dysfunction	(+)	(+)
NYHA class	II	III
BNP (pg/mL) (<18.4 pg/mL)	92.8	NA
NT-proBNP (pg/mL) (<125 pg/mL)	1,183	10,988
Hs-cTnT (ng/mL) (<0.014 ng/ml)	0.033	0.207
ECG		
Low QRS voltage	(+)	(+)
Pseudoinfarct pattern	(+)	(+)
Echocardiography		
LVEF (%) (>55%)	39	53
LAD (mm) (19–40 mm)	44	59
LVDd (mm) (39–55 mm)	57	NA
LVDs (mm) (22–42 mm)	46	NA
LVPWTd (mm) (8–12 mm)	12.5	16
IVSd (mm) (<12 mm)	13	20
Increased RV wall thickness	(+)	(−)
Restrictive LV filling pattern	(+)	(+)
Valvular abnormalities	(−)	(−)
Pericardial effusion	(−)	(−)
Autonomic dysfunction	(−)	(−)
Orthostatic hypotension	(−)	(−)
Gastrointestinal manifestations	(−)	(−)
Loss of weight	(−)	(−)
Polyneuropathy	(−)	(−)
CNS manifestations	(−)	(−)
Small fiber neuropathy	(+) *	(−)
Carpal tunnel syndrome	(−)	(−)
Biceps tendon rapture	(−)	(−)
Spinal canal stenosis	(+) *	(−)
Ocular manifestations	(−)	(−)
Chronic kidney disease	(−)	(+)
TTR concentration (22−40 mg/dL)	4 mg/dL	10 mg/dL
Proportion ratio variant TTR: wild TTRof amyloid in cardiac specimens	50%	variant TTR peptide detected **
Outcome	Survival	Death

Abbreviation: NA, Not available; NYHA, New York Heart Association; BNP, B-type natriuretic peptide; Hs-cTnT, high-sensitivity cardiac troponin test; IVSd, interventricular septum diameter; LVDd, left ventricular diastolic diameter; LVDs, left ventricular systolic diameter; LVEF, left ventricular ejection fraction; LVPWd, left ventricular posterior wall diameter; CNS, central nervous system. * Patient 1 has the symptom of sensory disturbance in foot. ** LC-MS/MS and laser microdissection data in Appendix A.

## Data Availability

All data have been included in the manuscript. Requests for additional information and/or reagents should be addressed to the corresponding author.

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
