# Peer review of "Clinical and Biochemical Characterization of Hereditary ATTR Amyloidosis Caused by a Novel Transthyretin Variant V121A (p.V141A)"

_ijms, 2025, doi:10.3390/ijms26104659_

Round 1
Reviewer 1 Report
Comments and Suggestions for Authors
Over 150 transthyretin (TTR) mutations have been identified in hereditary transthyretin (ATTRv) amyloidosis, and the paper by Yoshinaga et al. reports the identification of yet another novel TTR variant, namely V121A, identified in two unrelated amyloidosis patients aged > 60 years who developed cardiomyopathy of which, one died in less than four months. The topic is of great interest to researchers working on this not so rare disease. Results are sound and I have no reservation to its publication provided the comments below are addressed.
Page 2, line 64-64. The authors list small-molecule ligands, such as diflunisal, tafamidis, and acoramidis, but they forget tolcapone (Sant'Anna, R., Gallego, P., Robinson, L. et al. Repositioning tolcapone as a potent inhibitor of transthyretin amyloidogenesis and associated cellular toxicity. Nat Commun 7, 10787 (2016). https://doi.org/10.1038/ncomms10787) and its derivatives (Poonsiri, T.; Dell’Accantera, D.; Loconte, V.; Casnati, A.; Cervoni, L.; Arcovito, A.; Benini, S.; Ferrari, A.; Cipolloni, M.; Cacioni, E.; et al. 3-O-Methyltolcapone and Its Lipophilic Analogues Are Potent Inhibitors of Transthyretin Amyloidogenesis with High Permeability and Low Toxicity. Int. J. Mol. Sci. 2024, 25, 479. https://doi.org/10.3390/ijms25010479) which have been studied in the latest years and which represent the new frontier in treating hereditary transthyretin (ATTRv) amyloidosis.
Page 4, line 130. Figure 2, caption. “anti-TTR antibody (III)” should be “anti-TTR antibody (II)”
Page 9, line 291.” X-ray crystallographic structure of V121A-TTR.”. This is not an X-ray structure but rather a 3D model of V121A-TTR based on the X-ray crystallographic structure WT-TTR (PDB code 1F41). The caption should be changed to “3D model of V121A-TTR based on the X-ray crystallographic structure WT-TTR (PDB code 1F41) (Andreas Hörnberg, Therese Eneqvist, Anders Olofsson, Erik Lundgren, A.Elisabeth Sauer-Eriksson, A comparative analysis of 23 structures of the amyloidogenic protein transthyretin11Edited by F. Cohen, Journal of Molecular Biology, Volume 302, Issue 3, 2000, Pages 649-669,https://doi.org/10.1006/jmbi.2000.4078.)
Page 9, line 293. “Stable TTR is prone to tetramer formation compared to the dimeric mutants”. This comment is totally unrelated to the caption. Please report in the discussion and expand with references.
Reviewer 2 Report
Comments and Suggestions for Authors
The study identifies and characterizes a previously unreported transthyretin (TTR) variant (V121A), contributing to the understanding of ATTR amyloidosis. This discovery is significant given the increasing number of mutations linked to this disease. The authors utilized a comprehensive methodology, including clinical evaluations, biochemical assays, in silico modeling, structural analysis, and aggregation studies, to establish the pathogenicity of the V121A variant.
The findings contribute substantially to the knowledge of ATTR amyloidosis caused by the novel V121A TTR variant through detailed biochemical and clinical characterization. However, limitations such as a small sample size, absence of longitudinal data, and insufficient exploration of therapeutic options temper the results. Future research should prioritize larger cohorts, diverse populations, and deeper investigations into molecular mechanisms and treatment strategies to validate and expand upon these conclusions.
Comments and Suggestions for Improvement:
- Supplementary Figures: Please provide precise references in the main text to supplementary figures by numbering them and offering detailed descriptions.
- Patient and Methods Section:
- Line 135: The HGMD link (http://www.hgmd.org/) appears invalid or inaccessible.
- Line 136: The correct hyperlink is https://www.ncbi.nlm.nih.gov/clinvar/.
- Line 138: Update the hyperlink to https://gnomad.broadinstitute.org/.
- Lines 143–149: Consider removing these lines.
- Lines 155–160: Clarify how many recombinant TTR proteins were expressed, purified, and used in experiments. Specify whether they contained a His-tag or if it was mutated or cleaved. Provide a list of proteins used. Additionally, clarify whether the protein referred to as V121A (p.V141A) included the signal sequence.
- Line 229–249: Include references for the IP-MS experiment. A possible citation is:
Heegaard NH et al., J Sep Sci., 2006;29(3):371-377. doi:10.1002/jssc.200500377. - Also, confirm that the Xterra column is from Waters rather than Agilent.
- Results Section:
- Line 269: Specify which part of the supplemental file this refers to.
- Alignment of TTR protein sequences across species would help highlight conservation of the Val121 residue.
- Figure 3A: Clarify whether this represents an actual X-ray structure or a simulation/prediction.
- Figure 3D: Provide detailed descriptions of the five peaks.
- Line 318: Indicate which protein was used for aggregation studies and whether the His-tag influenced folding or aggregation.
- Line 338–348 and Figure 5:
Improve figure resolution; ensure labels on peaks are legible. - Verify mutations using alternative methods to rule out modifications such as methylation or oxidation potentially causing the same mass differences.
- Supplementary mass spectra require higher resolution for proper examination. Include details about sample preparation and instrumentation used for analysis.
These adjustments will enhance clarity, reproducibility, and overall quality of the article's findings.
Round 2
Reviewer 2 Report
Comments and Suggestions for Authors
I appreciate your responses to my questions.
I approve the revised manuscript, subject to the following minor changes:
1. Fig 5: V30L The wild-type (WT) peak is shown in green, while the mutant peak is displayed in red. In other cases, the color scheme is reversed. Is there a specific reason for this variation?
2. Supplementary Information (word file)
(All this information should be added at the beginning of the other supplementary PDF file.)
Question: Could you provide the specific proteomic database search parameters used to identify the modified peptide, given its absence from the current UniProt human database?
